# Abstraction based Output Range Analysis for Neural Networks

**Pavithra Prabhakar**[*],    **Zahra Rahimi Afzal**[*]
Department of Computer Science
Kansas State University
Manhattan, KS 66506
{pprabhakar,zrahimi}@ksu.edu

## Abstract

In this paper, we consider the problem of output range analysis for feed-forward neural networks with *ReLU* activation functions. The existing approaches reduce the output range analysis problem to satisfiability and optimization solving, which are NP-hard problems, and whose computational complexity increases with the number of neurons in the network. To tackle the computational complexity, we present a novel abstraction technique that constructs a simpler neural network with fewer neurons, albeit with interval weights called interval neural network (*INN*), which over-approximates the output range of the given neural network. We reduce the output range analysis on the *INN*s to solving a mixed integer linear programming problem. Our experimental results highlight the trade-off between the computation time and the precision of the computed output range.

## 1   Introduction

Neural networks are extensively used today in safety critical control systems such as autonomous vehicles and airborne collision avoidance systems [1, 16, 17, 18]. Hence, rigorous methods to ensure correct functioning of neural network controlled systems is imperative. Formal verification refers to a broad class of techniques that provide strong guarantees of correctness by exhibiting a proof. Formal verification of neural networks has attracted a lot of attention in the recent years [2, 5, 18, 28, 29, 31]. However, verifying neural networks is extremely challenging due to the large state-space, and the presence of nonlinear activation functions, and the verification problem is known to be NP-hard for even simple properties [18].

Our broad objective is to investigate techniques to verify neural network controlled physical systems such as autonomous vehicles. These systems consist of a physical system and a neural network controller that are connected in a feedback, that is, the output of the neural network is the control input (actuator values) to the physical system and the output of the physical system (sensor values) is input to the neural network controller. An important verification problem is that of safety, wherein, one seeks to ensure that the state of the neural network controlled system never reaches an unsafe set of states. This is established by computing the reachable set, the set of states reached by the system, and ensuring that the reach set does not intersect the unsafe states. An important primitive towards computing the reachable set is to compute the output range of a neural network controller given a set of input valuations.

In this paper, we focus on neural networks with rectified linear unit (*ReLU*) function as an activation function, and we investigate the output range computation problem for feed-forward neural networks [5]. Recently, there have been several efforts to address this problem that rely on satisfiability

---

[*]Both authors contributed equally to this work.

checking and optimization. Reluplex [18] is a tool that develops a satisfiability modulo theory for verifying neural networks, in particular, it encodes the input/output relations of a neural network as a satisfiability checking problem. A mixed integer linear programming (*MILP*) based approach is proposed in [5, 7] to compute the output range. These approaches construct constraints that encode the neural network behavior, and check satisfiability or compute optimal values over the constraints. The complexity of verification depends on the size of the constraints which in turn depends on the number of neurons in the neural network.

To increase the verification efficiency, we present an orthogonal approach that consists of a novel abstraction procedure to reduce the state-space (number of neurons) of the neural network. Abstraction is a formal verification technique that refers to methods for reducing the state-space while providing formal guarantees of properties that are preserved by the reduction. One of the well-studied abstraction procedures is predicate abstraction [4, 12] that consists of partitioning the state-space of a given system into a finite number of regions, and constructing an abstract system that consists of these regions as the states. Predicate abstraction has been employed extensively for safety verification, since, the safety of the abstract system is *sound*, that is, it implies the safety of the given system. Our main result consists of a *sound* abstraction that in particular over-approximates the output range of a given neural network. Note that an over-approximation can still provide useful safety analysis verdicts, since, if a superset of the reachable set does not intersect with the unsafe set, then the actual reachable set will also not intersect the unsafe set. The abstraction procedure essentially merges sets of neurons within a particular layer, and annotates the edges and biases with interval weights to account for the merging. Hence, we obtain a neural network with interval weights, which we call *interval neural networks* (*INN*s). While interval neural networks are more general than neural networks, we show as a proof of concept that the satisfiability and optimization based verification approaches can be extended to *INN*s by extending the *MILP* based encoding in [5] for neural networks to interval neural networks and use it to compute the output range of the abstract *INN*. We believe that other methods such as Reluplex can be extended to handle interval neural network, and hence, the abstraction procedure presented here can be used to reduce the state-space before applying existing or new verification algorithms for neural networks. An abstract interpretation based method has been explored in [11], wherein an abstract reachable set is propagated. However, our approach has the flavor of predicate abstraction [12] and computes an over-approximate system which can then be used to compute an over-approximation of the output range using any of the above methods including the one based on abstract interpretation [11].

The crucial part of the abstraction construction consists of appropriately instantiating the weights of the abstract edges. In particular, a convex hull of the weights associated with the concrete edges corresponding to an abstract edge does not guarantee soundness, which is shown using a counterexample in Section 3.1. We need to multiply the convex hull by a factor equivalent to the number of merged nodes in the source abstract node. The proof of soundness is rather involved, since, there is no straightforward relation between the concrete and the abstract states. We establish such a connection, by associating a set of abstract valuations with a concrete valuation for a particular layer, wherein, the abstract valuation for an abstract node takes values in the range given by the concrete valuations for the related concrete nodes. The crux of the proof lies in the observation (Proposition 1) that the behavior of a concrete valuation is mimicked in the abstract valuation by an average of the concrete valuations at the nodes corresponding to an abstract node. We conclude that the input/output relation associated with a certain layer of the concrete system is over-approximated by input/output valuations of the corresponding layer in the abstract system.

We have implemented our algorithm in a Python toolbox. We perform experimental analysis on the ACAS [15] case study, and observe that the verification time increases with the increase in the number of abstract nodes, however, the over-approximation in the output range decreases. Further, we notice that the output range can vary non-trivially even for a fixed number of abstract nodes, but different partitioning of the concrete nodes for merging. This suggests that further research needs to be done to understand the best strategies for partitioning the state-space of neurons for merging, which we intend to explore in the future.

**Related work.** Recent studies [2, 10, 14, 21, 29, 30] compare several neural network verification algorithms. Formal verification of feedforward neural networks with different activation functions have been considered. For instance, [11, 18] consider *ReLU*, where as [22, 23] consider large class of activation functions that can be represented as Lipschitz-continuous functions. We focus on *ReLU*

functions, but our method can be extended to more general functions. Different verification problems have been considered including output range analysis [6, 8, 14, 22, 26, 27], and robustness analysis [11, 20]. Verification methods include those based on reduction to satisfiability solving [10, 14, 18], optimizaiton solving [9], abstract interpretation [23, 24], and linearization [10, 21]. There is some recent work on verification of AI controlled cyber-physical systems [13, 25]

## 2   Interval Neural Network

A neural network (*NN*) is a computational model that consists of nodes (neurons) that are organized in layers and edges which are the connections between the nodes labeled by weights. An *NN* contains an input layer, some hidden layers, and an output layer each composed of neurons. Given values to the nodes in the input layer, the values at the nodes in the next layer are computed through a weighted sum dictated by the edge weights and the addition of the bias associated with the output node followed by an activation operation which we will assume is the *ReLU* (rectifier linear unit) function. In this section, we introduce interval neural networks *INN* that generalize neural networks with interval weights on edges and biases and will represent our abstract systems.

**Preliminaries.**   Let $\mathbb{R}$ denote the set of real numbers. Given a non-negative integer $k$, let $[k]$ denote the set $\{0, 1, \cdots, k\}$. Given a set $A$, $|A|$ represents the number of elements of $A$. For any two functions $f, g : A \to \mathbb{R}$, we say $f \leq g$ if $\forall s \in A, f(s) \leq g(s)$. We denote the *ReLU* function by $\sigma$, which is defined as $\sigma(x) = \max(0, x)$. Given two binary relations $R_1 \subseteq A \times B$ and $R_2 \subseteq B \times C$, we define their composition, denoted by $R_1 \circ R_2$, to be $\{(u, v) \mid \exists w, (u, w) \in R_1 \text{ and } (w, v) \in R_2\}$. For any set $S$, a valuation over $S$ is a function $f : S \to \mathbb{R}$. We define *Val(S)* to be the set of all valuations over $S$. A partition $R$ of the set $A$ is a set $R = \{R_1, ..., R_k\}$ such that $\bigcup_{i=1}^{k} R_i = A$ and $R_i \cap R_j = \emptyset$   $\forall i, j \in \{1, .., k\}$ and $i \neq j$.

**Definition 1** (Interval Neural Network). *An interval neural network (INN) is a tuple $(k, \{S_i\}_{i \in [k]},$ $\{W_i^l, W_i^u\}_{i \in [k-1]}, \{b_i^l, b_i^u\}_{i \in [k]/\{0\}})$, where*

- *$k$ is a natural number which we refer to as the number of layers;*

- *$\forall i \in [k]$, $S_i$ is a set of nodes of $i$-th layer in the interval neural network such that $\forall i \neq j$, $S_i \cap S_j = \emptyset$. $S_0$ is the input layer, $S_k$ is the output layer and $S_i$, $\forall i \in [k]/\{0, k\}$ is a hidden layer;*

- *$\forall i \in [k-1]$, $W_i^l, W_i^u : S_i \times S_{i+1} \to \mathbb{R}$ represent the weights of the edges between the $i$-th and $i+1$-th layer. We assume that $\forall i \in [k-1], s_i \in S_i, s_{i+1} \in S_{i+1}, W_i^l(s_i, s_{i+1}) \leq W_i^u(s_i, s_{i+1})$;*

- *$\forall i \in [k]/\{0\}$, $b_i^l, b_i^u : S_i \to \mathbb{R}$ are the biases associated with the nodes in the $i$-th layer, that is, $\forall i \in [k]/\{0\}, s_i \in S_i, b_i^l(s_i) \leq b_i^u(s_i)$.*

A neural network can be defined as a special kind of *INN* where the weights and biases are singular intervals.

**Definition 2** (Neural Network). *An INN $\mathcal{T}$ is a neural network (NN) if $\forall i \in [k-1], s \in S_i, s' \in S_{i+1}, W_i^l(s, s') = W_i^u(s, s')$, and $\forall i \in [k]/\{0\}, s \in S_i, b_i^l(s) = b_i^u(s)$.*

Figure 1 shows a neural network with 3 layers. The input layer has 2 nodes, the output layer has

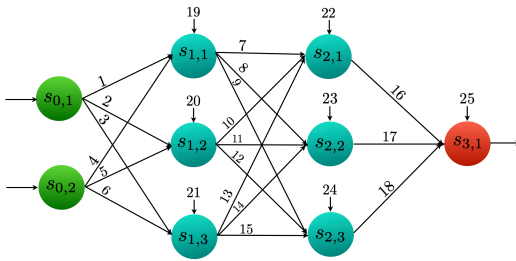

Figure 1: A neural network

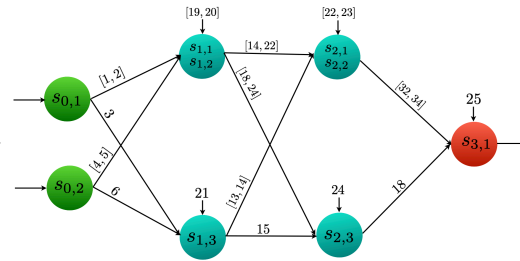

Figure 2: An interval neural network

1 node, and each of the hidden layers has 3 nodes. The weights on the edges are a single number

(singular intervals), hence, it is a neural network. Figure 2 shows an interval neural network again with 3 layers. The input and output layers have the same number of nodes as before, but the hidden layers have 2 nodes each. The weights on the edges are intervals (and non-singular), so this is an interval neural network (rather than just a neural network).

An execution of the neural network starts with valuations to the input nodes, and the valuations to the nodes of a certain layer are computed based on the valuations for the nodes in the previous layer. More precisely, to compute the value at a node $s_{i,j}$ corresponding to the $j$-th node in the layer $i$, we choose a weight from the interval for each of the incoming nodes and compute a weighted sum of the valuations of the nodes in the previous layer. Then a bias is chosen from the bias interval associated with $s_{i,j}$ and added to the weighted sum. Finally, the *ReLU* function is applied on this sum. The execution then proceeds to the next layer. The semantics of the neural network is captured using a set of pairs of input-output valuations wherein the output valuation is a possible result starting from the input valuation and executing the neural network. Next, we define the semantics of an *INN* as a set of valuations for the input and output layers.

**Definition 3** (Semantics of *INN* Network). *Given an INN* $\mathcal{T} = (k, \{S_i\}_{i \in [k]}, \{W_i^l, W_i^u\}_{i \in [k-1]}, \{b_i^l, b_i^u\}_{i \in [k]/\{0\}})$ *and* $i \in [k-1]$, $[|T|]_i = \{(v_1, v_2) \in \text{Val}(S_i) \times \text{Val}(S_{i+1}) \mid \forall s' \in S_{i+1}, v_2(s') = \sigma(\sum_{s \in S_i} w_{s,s'} v_1(s) + b_{s'}),$ *where* $W_i^l(s, s') \le w_{s,s'} \le W_i^u(s, s')$, $b_i^l(s') \le b_{s'} \le b_i^u(s')\}$. *We define* $[|T|] = [|T|]_0 \circ [|T|]_1 \circ \cdots \circ [|T|]_{k-1}$.

The semantics can be captured alternately using a post operator, that given a valuation of layer $i$, returns the set of all valuations of layer $i + 1$ that are consistent with the semantics.

**Definition 4.** *Given an INN* $\mathcal{T}$ *with k layers,* $i \in [k-1]$ *and* $V \subseteq \text{Val}(S_i)$, *we define* $\text{Post}_{\mathcal{T},i}(V) = \{v' \mid \exists v \in V, (v, v') \in [|T|]_i\}$. *Given* $V \subseteq \text{Val}(S_0)$, *we define* $\text{Post}_{\mathcal{T}}(V) = \{v' \mid \exists v \in V, (v, v') \in [|T|]\}$.

For notational convenience, we will write $\text{Post}_{\mathcal{T},i}(\{v\})$ and $\text{Post}_{\mathcal{T}}(\{v\})$ as just $\text{Post}_{\mathcal{T},i}(v)$ and $\text{Post}_{\mathcal{T}}(v)$, respectively.

Our objective is to find an over-approximation of the values the output neurons can take in an interval neural network, given a set of valuations for the input layer.

**Problem 1** (Output range analysis). *Given an INN* $\mathcal{T}$ *with k layers and a set of input valuations* $I \subseteq \text{Val}(S_0)$, *compute valuations* $l, u \in \text{Val}(S_k)$ *such that* $\forall (v_1, v_2) \in [|T|]$ *if* $v_1 \in I$ *then* $l(s) \le v_2(s) \le u(s)$ *for every* $s \in S_k$.

## 3 Our Approach

In this section, we present an abstraction based approach for over-approximating the output range of an interval neural network. First, in Section 3.1, we describe the construction of an abstract system whose semantics over-approximates the semantics of a given *INN* and argue the correctness of the construction. In Section 3.2, we present an encoding of the interval neural network to mixed integer linear programming that enables the computation of the output range.

### 3.1 Abstraction of an *INN*

The motivation for the abstraction of an *INN* is to reduce the "state-space", the number of neurons in the network, so that computation of the output range can scale to larger *INN*s. Our broad idea consists of merging the nodes of a given concrete *INN* so as to construct a smaller abstract *INN*. However, it is crucial that we instantiate the weights on the edges and the biases appropriately to ensure that the semantics of the abstracted system is an over-approximation of the concrete *INN*. For instance, consider the neural network in Figure 3 and consider an input value 1. It results in an output value of 2. Figure 4 abstracts the neural network in Figure 3 by taking the convex hull of the weights on the concrete edges corresponding to the abstract edge. However, given input 1, the output of the abstract neural network is 1 and does not contain 2. Hence, we need to be careful in the construction of the abstract system.

Given two sets of concrete nodes from consecutive layers of the *INN*, $\hat{s}_1$ and $\hat{s}_2$, which are each merged into one abstract node, we associate an interval with the edge between $\hat{s}_1$ and $\hat{s}_2$ to be the interval $[|\hat{s}_1|w_1, |\hat{s}_1|w_2]$, where $w_1$ and $w_2$ are the minimum and maximum weights associated with the edges in the concrete system between nodes in $\hat{s}_1$ and $\hat{s}_2$, respectively, and $|\hat{s}_1|$ is the number

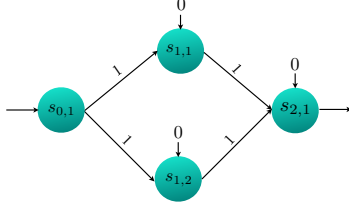

Figure 3: A concrete neural network

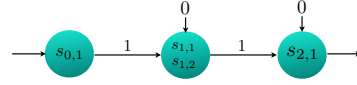

Figure 4: An incorrect abstraction

of concrete nodes corresponding to the abstract node $\hat{s}_1$. In other words, $[w_1, w_2]$ is the convex hull of the intervals associated with the edges between nodes in $\hat{s}_1$ and $\hat{s}_2$ multiplied by a factor corresponding to the number of concrete nodes corresponding to the source abstract node. Note that the above abstraction will lead to a weight of 2 on the second edge in Figure 4, thus leading to an output of 2 as in the concrete system.

Next, we formally define the abstraction. We say that $P = \{P_i\}_{i \in [k]}$ is a partition of $\mathcal{T}$, if for every $i$, $P_i$ is a partition of the $S_i$, the nodes in the $i$-th layer of $\mathcal{T}$.

**Definition 5** (Abstract Neural Network)**.** *Given an INN* $T = (k, \{S_i\}_{i \in [k]}, \{W_i^l, W_i^u\}_{i \in [k-1]}, \{b_i^l, b_i^u\}_{i \in [k]/\{0\}})$ *and a partition* $P = \{P_i\}_{i \in [k]}$ *of* $T$, *we define an INN* $T/P = (k, \{\hat{S}_i\}_{i \in [k]}, \{\widehat{W}_i^l, \widehat{W}_i^u\}_{i \in [k-1]}, \{\hat{b}_i^l, \hat{b}_i^u\}_{i \in [k]/\{0\}})$, *where*

- $\forall i \in [k]$, $\hat{S}_i = P_i$;

- $\forall i \in [k-1]$, $\hat{s}_i \in \hat{S}_i$, $\hat{s}_{i+1} \in \hat{S}_{i+1}$, $\widehat{W}_i^l(\hat{s}_i, \hat{s}_{i+1}) = |\hat{s}_i| \min \{W_i^l(s_i, s_{i+1}) \mid s_i \in \hat{s}_i, \ s_{i+1} \in \hat{s}_{i+1}\}$ *and* $\widehat{W}_i^u(\hat{s}_i, \hat{s}_{i+1}) = |\hat{s}_i| \max \{W_i^u(s_i, s_{i+1}) \mid s_i \in \hat{s}_i, \ s_{i+1} \in \hat{s}_{i+1}\}$;

- $\forall i \in [k]/\{0\}$, $\hat{s}_i \in \hat{S}_i$, $\hat{b}_i^l(\hat{s}_i) = \min \{b_i^l(s_i) \mid s_i \in \hat{s}_i\}$ *and* $\hat{b}_i^u(\hat{s}_i) = \max \{b_i^u(s_i) \mid s_i \in \hat{s}_i\}$.

Figure 2 shows the abstraction of the neural network in Figure 1, where the nodes $s_{1,1}$ and $s_{1,2}$ are merged and the nodes $s_{2,1}$ and $s_{2,2}$ are merged. Note that the edge from $\{s_{1,1}, s_{1,2}\}$ to $\{s_{2,1}, s_{2,2}\}$ has weight interval $[14, 22]$, which is obtained by taking the convex hull of the four weights $7, 10, 8$ and $11$, and multiplying by 2, the size of the source abstract node.

The following theorem states the correctness of the construction of $T/P$. It states that every input/output valuation that is admitted by $T$ is also admitted by $T/P$, thus establishing the soundness of the abstraction.

**Theorem 1.** *Given an INN* $T = (k, \{S_i\}_{i \in [k]}, \{W_i^l, W_i^u\}_{i \in [k-1]}, \{b_i^l, b_i^u\}_{i \in [k]/\{0\}})$ *and a partition* $P = \{P_i\}_{i \in [k]}$ *of* $T$ *such that* $P_0 = S_0$ *and* $P_k = S_k$, $[|T|] \subseteq [|T/P|]$.

We devote the rest of the section to sketch a proof of Theorem 1. Broadly, the proof consists of relating the valuations in the $i$-th layer of the concrete *INN* with the $i$-th layer of the abstract *INN*. Note that the nodes in a particular layer of the abstract and the concrete system might not be the same. The following definition relates states in the concrete system to those in the abstract system.

**Definition 6.** *Given a valuation* $v \in Val(S_i)$, $AV(v) = \{\hat{v} \in Val(\hat{S}_i) \mid \forall \hat{s} \in \hat{S}_i, \min_{s \in \hat{s}} v(s) \leq \hat{v}(\hat{s}) \leq \max_{s \in \hat{s}} v(s)\}$.

Given a valuation $v$ of the $i$-th layer of the concrete system, $AV(v)$ consists of the set of all abstract valuations of the $i$-th layer in the abstract system, where each abstract node gets a value which is within the range of values of the corresponding concrete nodes. Proof of Theorem 1 relies on the following connection between corresponding layers of the concrete and abstract *INN*s.

**Lemma 1.** *If* $(v, v') \in [|T|]_i$, *then* $AV(v') \subseteq Post_{T/P,i}(AV(v))$.

The proof of Lemma 1 broadly follows the following structure. We first observe that the abstraction procedure corresponding to edges between layer $i$ and layer $i + 1$ can be decomposed into two steps, wherein we first merge the nodes of the $i$-th layer and then we merge the nodes of the $i + 1$-st layer. Note that

$$\widehat{W}_i^l(\hat{s}_i, \hat{s}_{i+1}) = |\hat{s}_i| \min_{s_i \in \hat{s}_i, \ s_{i+1} \in \hat{s}_{i+1}} W_i^l(s_i, s_{i+1}) = \min_{s_{i+1} \in \hat{s}_{i+1}} |\hat{s}_i| \min_{s_i \in \hat{s}_i} W_i^l(s_i, s_{i+1})$$

A similar observation can be made about the max. Hence, our first step consists of a function *labs* which merges the nodes in the "left" layer and associates an interval with the edges which corresponds to computing the convex hull followed by multiplying with an appropriate factor. Next, the function *rabs* merges the nodes in the "right" layer and associates an interval which corresponds to only computing the convex hull. Next, we define these abstraction functions, and state their relation with the concrete systems.

**Definition 7.** *Given an INN* $T = (k, \{S_i\}_{i \in [k]}, \{W_i^l, W_i^u\}_{i \in [k-1]}, \{b_i^l, b_i^u\}_{i \in [k]/\{0\}})$, $j$ *and* $P$ *which is a partition of jth layer of* $\mathcal{T}$, *we define an INN* $labs(\mathcal{T}, j, P) = (1, \{\hat{S}_i\}_{i \in [1]}, \{\hat{W}_i^l, \hat{W}_i^u\}_{i \in [0]}, \{\hat{b}_i^l, \hat{b}_i^u\}_{i \in \{1\}})$, *where*

- $\hat{S}_0 = P$, $\hat{S}_1 = S_{j+1}$;

- $\forall \hat{s}_0 \in \hat{S}_0, \hat{s}_1 \in \hat{S}_1, \widehat{W}_0^l(\hat{s}_0, \hat{s}_1) = |\hat{s}_0| \min \{W_0^l(s_0, \hat{s}_1) \mid s_0 \in \hat{s}_0\}$, *and* $\widehat{W}_0^u(\hat{s}_0, \hat{s}_1) = |\hat{s}_0| \max \{W_0^u(s_0, \hat{s}_1) \mid s_0 \in \hat{s}_0\}$;

- $\forall \hat{s}_1 \in \hat{S}_1, \hat{b}_1^l(\hat{s}_1) = b_1^l(\hat{s}_1)$, *and* $\hat{b}_1^u(\hat{s}_1) = b_1^u(\hat{s}_1)$.

Figure 5 show the left abstraction of the the neural network in Figure 1 with respect to layer 1, where

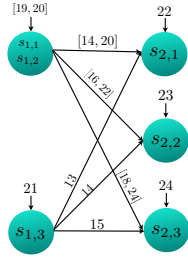

Figure 5: A left abstraction illustration

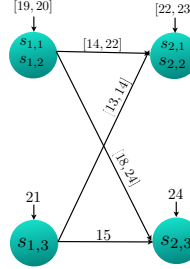

Figure 6: A right abstraction illustration

the nodes $s_{1,1}$ and $s_{1,2}$ are merged. The edge from $\{s_{1,1}, s_{1,2}\}$ to $s_{2,1}$ has weight $[14, 20]$ which is obtained by taking the convex hull of the values 7 and 10 and multiplying by 2.

**Definition 8.** *Given an INN* $\mathcal{T} = (1, \{S_i\}_{i \in [1]}, \{W_i^l, W_i^u\}_{i \in [0]}, \{b_i^l, b_i^u\}_{i \in \{1\}})$ *and* $P$ *which is a partition of the layer* 1 *of* $\mathcal{T}$, *we define an INN* $rabs(\mathcal{T}, P) = (1, \{\hat{S}_i\}_{i \in [1]}, \{\hat{W}_i^l, \hat{W}_i^u\}_{i \in [0]}, \{\hat{b}_i^l, \hat{b}_i^u\}_{i \in \{1\}})$, *where*

- $\hat{S}_0 = S_0$, $\hat{S}_1 = P$;

- $\forall \hat{s}_0 \in \hat{S}_0, \hat{s}_1 \in \hat{S}_1, \widehat{W}_0^l(\hat{s}_0, \hat{s}_1) = \min \{W_0^l(\hat{s}_0, s_1) \mid s_1 \in \hat{s}_1\}$, *and* $\widehat{W}_0^u(\hat{s}_0, \hat{s}_1) = \max \{W_0^u(\hat{s}_0, s_1) \mid s_1 \in \hat{s}_1\}$;

- $\forall \hat{s}_1 \in \hat{S}_1, \hat{b}_1^l(\hat{s}_1) = \min \{b_1^l(s_1) \mid s_1 \in \hat{s}_1\}$, *and* $\hat{b}_1^u(\hat{s}_1) = \max \{b_1^u(s_1) \mid s_1 \in \hat{s}_1\}$.

Figure 6 shows the right abstraction of the interval neural network in Figure 5, where the nodes $s_{2,1}$ and $s_{2,2}$ are merged. The edge from $\{s_{1,1}, s_{1,2}\}$ to $\{s_{2,1}, s_{2,2}\}$ has weight $[14, 22]$ which is obtained by taking the convex hull of the intervals $[14, 20]$ and $[16, 22]$. Note that Figure 6 is the same as the Figure 2 with restricted 2 layers 1 and 2.

Note that applying the left abstraction followed by right abstraction to the $j$-th layer of $\mathcal{T}$ gives us the $j$-th layer of $T/P$. This is stated in the following lemma.

**Lemma 2.** $rabs(labs(\mathcal{T}, j, P_j), P_{j+1}) = (1, \{\hat{S}_{i+j}\}_{i \in [1]}, \{\hat{W}_{i+j}^l, \hat{W}_{i+j}^u\}_{i \in [0]}, \{\hat{b}_{i+j}^l, \hat{b}_{i+j}^u\}_{i \in \{1\}})$.

Proof of Lemma 1 relies on some crucial properties which we state below. The crux of the proof of the correctness of left abstraction lies in the following proposition. It states that the contribution of the values of a set of left nodes $\{s_1, \cdots, s_n\}$ on a right node $s$, can be simulated in a left abstraction which merges $\{s_1, \cdots, s_n\}$ by the average of the values.

**Proposition 1.** *Let* $v_1, v_2, \ldots, v_n$ *and* $w_1, w_2, \ldots, w_n$ *be real numbers. Let* $\bar{v} = \sum_i v_i/n$. *There exists a* $w$ *such that* $n \min_i w_i \leq w \leq n \max_i w_i$ *and* $\sum_i w_i v_i = \bar{v} w$.

**Proposition 2.** *If $(v_1, v_2) \in [[T]]_i$, then $v_2 \in Post_{labs(\mathcal{T},j,P)}(AV(v_1))$.*

Next, we state the correctness of *rabs*. Here we show that given any valuation $v$ of the right layer in the concrete system, any valuation $\hat{v} \in AV(v)$ can be obtained in the abstraction. It relies on the observation that $\hat{v}(\hat{s})$ is a convex combination of the $\min_{s \in \hat{s}} v(s)$ and $\min_{s \in \hat{s}} v(s)$, and the weight interval of an abstract edge is a convex hull of the intervals of the corresponding concrete edges.

**Proposition 3.** *Given an INN $\mathcal{T}$ with one layer, and a partition $P$ of layer 1, if $(v_1, v_2) \in [[T]]$, then $AV(v_2) \subseteq Post_{rabs(\mathcal{T},P)}(v_1)$.*

Proofs are eliminated due to shortage of space and are provided in the supplementary material.

### 3.2 Encoding the interval neural network and *MILP* solver

In this section, we present a reduction of the range computation problem to solving a mixed integer linear program. The ideas are similar to those in [3, 19] using the big-M method. However, since, our weights on the edges are not unique but come from an interval, a direct application of the previous encodings where the constant weights are replaced by a variable with additional constraints related to the interval the weight variable is required to lie in, results in non-linear constraints. However, we observe that we can eliminate the weight variable by replacing it appropriately with the minimum and maximum values of the interval corresponding to it.

We encode the semantics of an *INN* $\mathcal{T}$ as a constraint $Enc(\mathcal{T})$ over the following variables. For every node $s$ of the *INN* $\mathcal{T}$, we have a real valued variable $x_s$, and we have a binary variable $q_s$ that takes values in $\{0, 1\}$. Let $X_i$ denote the set of variables $\{x_s \mid s \in S_i\}$, and $Q_i = \{q_s \mid s \in S_i\}$. Given a valuation $v \in Val(S_i)$, we will abuse notation and use $v$ to also denote a valuation of $X_i$, wherein $v$ assigns to $x_s \in X_i$, the valuation $v(s)$, and vice versa. Let $X = \cup_i X_i$ and $Q = \cup_i Q_i$. $Enc(\mathcal{T})$ is the union of the encodings of the different layers of $\mathcal{T}$, that is, $Enc(\mathcal{T}) = \cup_i Enc(\mathcal{T}, i)$, where $Enc(\mathcal{T}, i)$ denotes the constraints corresponding to layer $i$ of $\mathcal{T}$. $Enc(\mathcal{T}, i)$ in turn is the union of constraints corresponding to the different nodes in layer $i + 1$, that is, $Enc(\mathcal{T}, i) = \cup_{s' \in S_{i+1}} C_{s'}^{i+1}$, where the constraints in $C_{s'}^{i+1}$ are as below:

$$C_{s'}^{i+1} : \begin{cases} \sum_{s \in S_i} W_i^l(s, s')x_s + b_i^l(s') \leq x_{s'}, \ 0 \leq x_{s'} \\ \sum_{s \in S_i} W_i^u(s, s')x_s + b_i^u(s') + Mq_{s'} \geq x_{s'}, \ M(1 - q_{s'}) \geq x_{s'} \end{cases} \quad (1)$$

Here, $M$ is an upper bound on the absolute values any neuron can take (before applying the *ReLU* operation) for a given input set. It can be estimated using the norms of the weights interpreted as matrices, that is, $\|W_i^l\|$ and $\|W_i^u\|$, an interval box around the input polyhedron, and the norms of biases.

Next, we state and prove the correctness of the encoding. More precisely, we show that $(v, v') \in [[T]]_i$ if and only if there are valuations for the variables in $Q_i$ such that the constraints $Enc(T, i)$ are satisfied when the values for $X_i$ and $X_{i+1}$ are provided by $v$ and $v'$.

**Theorem 2.** *Let $v \in Val(S_i)$ and $v' \in Val(S_i)$. Then $(v, v') \in [[T]]_i$ if and only if there is a valuation $z \in Val(Q_i)$, such that $Enc(T, i)$ is satisfied with values $v, v'$ and $z$.*

We can now compute the output range analysis by solving a maximization and a minimization problem for each output variable. More precisely, for each $s \in S_k$, the output layer, we solve: $\max x_s$ such that $Enc(\mathcal{T})$ and *I* hold, where *I* is a constraint on the input variables encoding the set of input valuations. Similarly, we solve a minimization problem, and thus obtain an output range for the variable $x_s$ given the input set of valuations *I*. The maximization and minimization problems can be solved using mixed integer linear programming (*MILP*) if *I* is specified using linear constraints. Even checking satisfiability of a set of mixed integer linear constraints is NP-hard problems, however, there are commercial software tools that solve *MILP* such as Gurobi and CPLEX.

## 4 Implementation

In this section, we present our experimental analysis using a Python toolbox that implements the abstraction procedure and the reduction of the *INN* output range computation to *MILP* solving. We consider as a case study ACAS Xu benchmarks, which are neural networks with 6 hidden layer with

each layer consisting of 50 neurons [2]. We report here the results with one of the benchmarks, we observed similar behavior with several other benchmarks.

We consider abstractions of the benchmark with different number of abstract nodes, namely, 2, 4, 8, 16, 32, which are generated randomly. For a fixed number of abstract nodes, we perform 30 different random runs, and measure the average, maximum and minimum time for different parts of the analysis. Similarly, we compute the output range for a fixed number of abstract nodes, and obtain the average, maximum and minimum on the lower and upper bound of the output ranges. The lower bound was unanimously 0, hence, we do not report it here. The results are summarized in Figures 7, 8, 9, and 10.

As shown in Figure 7, the abstraction construction time increases gradually with the number of abstract neurons. We observe a similar trend with encoding time. However, the time taken by Gurobi to solve the *MILP* problems increases drastically after certain number of abstract nodes. Also, as shown in Figure 9, the *MILP* solving time by Gurobi is the most expensive part of the overall computation. Since, this is directly proportional to the number of abstract nodes, abstraction procedure proposed in the paper, has the potential to reduce the range computation time drastically. In fact, Gurobi did not return when ACAS Xu benchmark was encoded without any abstraction, thus, demonstrating the usefulness of the abstraction.

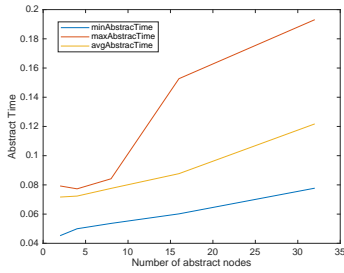

Figure 7: Abstraction Time

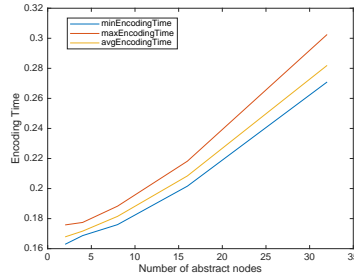

Figure 8: Encoding Time

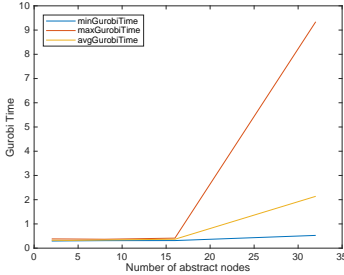

Figure 9: *MILP* Solving Time

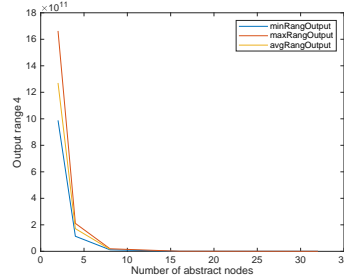

Figure 10: Output Range

We compare output ranges (upper bounds) based on different abstractions. The upper bound of the output range decreases as we consider more abstract nodes, since, the system becomes more precise. In fact, it decreases very drastically in the first few abstraction. We compute the average, minimum and maximum of the upper bound on the output range. Even for a fixed number of abstract nodes, the maximum and minimum of the upper bound on the output range among the random runs has a wide range, and depends on the specific partitioning. For instance, as seen in the Figure 10, although we have only 2 partitions, the upper bound on the output range varies by a factor of 2. This suggest that the partitioning strategy can play a crucial role in the precision of output range. Hence, we plan to explore partitioning strategies in the future. To conclude, our method provides a trade-off between verification time and the precision of the output range depending on the size of the abstraction.

## 5 Conclusions

In this paper, we investigated a novel abstraction techniques for reducing the state-space of neural networks by introducing the concept of interval neural networks. Our abstraction technique is orthogonal to existing techniques for analyzing neural networks. Our experimental results demonstrate the usefulness of abstraction procedure in computing the output range of the neural network, and the trade-off between the precision of the output range and the computation time. However, the precision of the output range is affected by the specific choice of the partition of the concrete nodes even for a fixed number of abstract nodes. Our future direction will consist of exploring different partition strategies for the abstraction with the aim of obtaining precise output ranges. In addition, we will consider more complex activation function. Our abstraction technique will extend in a straightforward manner, however, we will need to investigate methods for analyzing the "interval" version of the neural network for these new activation functions.

## Acknowledgments

Pavithra Prabhakar was partially supported by NSF CAREER Award No. 1552668 and ONR YIP Award No. N000141712577.

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
