[Supplementary Material]

# 1 Supplementary material

**Proof of Proposition 1.** Let $w = \sum_i w_i v_i / \bar{v}$. Then it trivially satisfies $\sum_i w_i v_i = \bar{v} w$. Let $w_{\min} = \min_i w_i$ and $w_{\max} = \max_i w_i$. We need to show that $n w_{\min} \leq w \leq n w_{\max}$. Note that $w = \sum_i w_i v_i / \bar{v} \geq \sum_i w_{\min} v_i / \bar{v} = w_{\min} \sum_i v_i / \bar{v} = w_{\min} n \bar{v} / \bar{v} = n w_{\min}$. Similarly, we can show that $w = \sum_i w_i v_i / \bar{v} \leq n w_{\max}$. The following proposition captures the relation between a layer of the concrete system and its left abstraction.

**Proof of Proposition 2.** Let $(v_1, v_2) \in [|T|]_i$. Let $S_0$ and $S_1$ be the left and right layers of $labs(\mathcal{T}, j, P)$, respectively. From the definition of the semantic of *INN* given by Definition 3, we know that for any $s' \in S_1$, $v_2(s') = \sigma(\sum_{s \in S_0} w_{s,s'} v_1(s) + b_{s'})$, where for every $s$, $W_0^l(s, s') \leq w_{s,s'} \leq W_0^u(s, s')$, $b_1^l(s') \leq b_{s'} \leq b_1^u(s')$. We can group together all neurons that are merged together in $P$, and rewrite the above as $v_2(s') = \sigma(\sum_{\hat{s} \in P} \sum_{s \in \hat{s}} w_{s,s'} v_1(s) + b_{s'})$. From Proposition 1, we can replace $\sum_{s \in \hat{s}} w_{s,s'} v_1(s)$ by $v_{\hat{s}} w_{\hat{s}}$, where $v_{\hat{s}} = \sum_{s \in \hat{s}} v_1(s) / |\hat{s}|$ and $w_{\hat{s}}$ is such that $|\hat{s}| \min_{s \in \hat{s}} w_{s,s'} \leq w_{\hat{s}} \leq |\hat{s}| \max_{s \in \hat{s}} w_{s,s'}$. Consider a valuation $\hat{v}_1$, where $\hat{v}_1(\hat{s}) = v_{\hat{s}} = \sum_{s \in \hat{s}} v_1(s) / |\hat{s}|$. Since, the average is in between the minimum and maximum values, $\hat{v}_1 \in AV(v_1)$. Now $v_2$ can be rewritten using $\hat{v}_1$ as $v_2(s') = \sigma(\sum_{\hat{s} \in P} v_{\hat{s}} w_{\hat{s}} + b_{s'}) = \sigma(\sum_{\hat{s} \in P} \hat{v}_1(\hat{s}) w_{\hat{s}} + b_{s'})$, where $|\hat{s}| \min_{s \in \hat{s}} w_{s,s'} \leq w_{\hat{s}} \leq |\hat{s}| \max_{s \in \hat{s}} w_{s,s'}$. Since $w_{s,s'}$ also satisfies $|\hat{s}| \min_{s \in \hat{s}} W_0^l(s, s') \leq w_{\hat{s}} \leq |\hat{s}| \max_{s \in \hat{s}} W_0^u(s, s')$, we see that $v_2 \in Post_{labs(\mathcal{T},j,P)}(AV(v_1))$ (since, $v_2$ and $\hat{v}_1$ satisfy the semantics of $labs(\mathcal{T}, j, P)$).

**Proof of Proposition 3.** Consider $\hat{v}_2 \in AV(v_2)$. Then $\hat{v}_2(\hat{s}') = \alpha v_2(s_1') + (1 - \alpha) v_2(s_2')$, where $s_1'$ is the node in $\hat{s}'$ for which $v_2$ at the node is the minimum and $s_2'$ is the node in $\hat{s}'$ for which $v_2$ at the node is the maximum. Let $S_0$ and $S_1$ be the nodes in the left and right layers of $\mathcal{T}$. $v_2(s_i') = \sigma(\sum_{s \in S_0} w_{s,s_i'} v_1(s) + b_{s_i'})$, where for every $s$, $W_0^l(s, s_i') \leq w_{s,s_i'} \leq W_0^u(s, s_i')$, $b_1^l(s_i') \leq b_{s_i'} \leq b_1^u(s_i')$. $\hat{v}_2(\hat{s}') = \alpha v_2(s_1') + (1 - \alpha) v_2(s_2') = \alpha \sigma(\sum_{s \in S_0} w_{s,s_1'} v_1(s) + b_{s_1'}) + (1 - \alpha) \sigma(\sum_{s \in S_0} w_{s,s_2'} v_1(s) + b_{s_2'})$. Let us first consider the case where the expressions within $\sigma$ are non-negative. Then $\hat{v}_2(\hat{s}') = \alpha(\sum_{s \in S_0} w_{s,s_1'} v_1(s) + b_{s_1'}) + (1 - \alpha)(\sum_{s \in S_0} w_{s,s_2'} v_1(s) + b_{s_2'}) = \sum_{s \in S_0} (\alpha w_{s,s_1'} + (1 - \alpha) w_{s,s_2'}) v_1(s) + (\alpha b_{s_1'} + (1 - \alpha) b_{s_2'}) = \sum_{s \in S_0} w_{s,s_1',s_2'} v_1(s) + b_{s_1',s_2'}$, where $w_{s,s_1',s_2'} = (\alpha w_{s,s_1'} + (1 - \alpha) w_{s,s_2'})$ and $b_{s_1',s_2'} = \alpha b_{s_1'} + (1 - \alpha) b_{s_2'}$. Note that $w_{s,s_1',s_2'}$ and $b_{s_1',s_2'}$ are in the edge weights and biases of the abstract system. If $\sum_{s \in S_0} w_{s,s_2'} v_1(s) + b_{s_2'}$ is negative, then $v_2(s_2') = v_2(s_1') = 0$, hence, $\hat{v}_2(\hat{s}') = 0$ can be simulated using either the values used to obtain $v_2(s_1')$ or $v_2(s_2')$. If $v_2(s_1') = 0$, but $v_2(s_2') > 0$, then we note that $\sum_{s \in S_0} w_{s,s_1'} v_1(s) + b_{s_1'} \leq 0$ and $(\sum_{s \in S_0} w_{s,s_2'} v_1(s) + b_{s_2'}) > 0$, any linear combination of the two can still be obtained using $v_1$, and $(1 - \alpha)(\sum_{s \in S_0} w_{s,s_2'} v_1(s) + b_{s_2'})$ is between the two values and can be obtained from $v_1$, and further, applying $\sigma$ would give us $v_2(s_2')$.

**Proof of Lemma 1.** Suppose $(v, v') \in [|T|]_i$. Then from Proposition 2, we have $v' \in Post_{labs(\mathcal{T},i,P_{i+1})}(AV(v))$, and further, from Proposition 3, we have $AV(v') \subseteq Post_{rabs(labs(\mathcal{T},i,P_{i+1}),P_i)}(AV(v))$. Finally, from Lemma 2, we obtain that $AV(v') \subseteq Post_{T/P,i}(AV(v))$.

**Proof of Theorem 1.** Suppose $(v, v') \in [|T|]$, then there exists a sequence of valuations $v_0, v_1, \cdots, v_k$, where $v_0 = v'$ and $v_i \in Post_{\mathcal{T},i}(v_{i-1})$ for $i > 0$. From Lemma 1, we know that since $(v_i, v_{i+1}) \in [|T|]_i$, $AV(v_{i+1}) \subseteq Post_{T/P,i}(AV(v_i))$. Since, $[|T|]$ is the composition of $[|T|]_i$, we obtain that $AV(v_k) \subseteq Post_{T/P}(AV(v_0))$. If the nodes in the input and output layer are not merged, then, $AV(v_0) = \{v_0\} = \{v\}$ and $AV(v_k) = \{v_k\} = \{v'\}$. Therefore, $(v, v') \in [|T|]$.

**Proof of Theorem 2.** First, let us prove that if $(v, v') \in [|T|]_i$, then there is a valuation $z \in Val(Q_i)$, such that $Enc(T, i)$ is satisfied with values $v, v'$ and $z$. In fact, it suffices to fix an $s'$ and show that $C_{s'}^{i+1}$ is satisfied by $v, v'(s')$ and $z(q_{s'})$. First, note that $v'(s') \geq 0$ since it is obtained by applying the *ReLU* function, so the second constraint in $C_{s'}^{i+1}$ is satisfied. From the semantics, we know that $v'(s') = \sigma(\sum_{s \in S_i} w_{s,s'} v(s) + b_{s'})$, where $W_i^l(s, s') \leq w_{s,s'} \leq W_i^u(s, s')$ and $b_i^l(s') \leq b_{s'} \leq b_i^u(s')$. Hence, $\sigma(\sum_{s \in S_i} W_i^l(s, s') v(s) + b_i^l(s')) \leq v'(s') \leq \sigma(\sum_{s \in S_i} W_i^u(s, s') v(s) + b_i^u(s'))$. Let $v''(s') = \sum_{s \in S_i} w_{s,s'} v(s) + b_{s'}$, that is, $v'(s') = \sigma(v''(s'))$.

Case $v''(s') \geq 0$: $v'(s') = v''(s')$ and we have $\sum_{s \in S_i} W_i^l(s, s')v(s) + b_i^l(s') \leq v'(s') \leq \sum_{s \in S_i} W_i^u(s, s')v(s) + b_i^u(s')$. Hence, for $z(q_{s'}) = 0$, the first, third and fourth constraints in $C_{s'}^{i+1}$ are satisfied.

Case $v''(s') < 0$: In this case, $v'(s') = 0$ and we set $z(q_{s'}) = 1$. $\sum_{s \in S_i} W_i^l(s, s')v(s) + b_i^l(s') \leq \sum_{s \in S_i} w_{s,s'}v(s) + b_{s'} = v''(s') < 0 = v'(s')$, so the first constraint is satisfied. $\sum_{s \in S_i} W_i^u(s, s')v(s) + b_i^u(s') + Mq_{s'} = \sum_{s \in S_i} W_i^u(s, s')v(s) + b_i^u(s') + M$. Since, $M$ is an upperbound on the absolute value of $x_{s'}$ before applying the *ReLU* operation, $\sum_{s \in S_i} W_i^u(s, s')v(s) + b_i^u(s') + M$ is positive, and hence, satisfies the third constraint. The fourth constraint is satisfied by the choice of $M$, that is, $M \geq x_{s'}$.

Next, we prove the other direction. Suppose $C_{s'}^{i+1}$ is satisfied for every $s'$ by some $v, v', z$, then we show that $(v, v') \in [\![T]\!]_i$.

Case $z(q_{s'}) = 0$: In this case, we have $\sum_{s \in S_i} W_i^l(s, s')x_s + b_i^l(s') \leq x_{s'} \leq \sum_{s \in S_i} W_i^u(s, s')x_s + b_i^u(s')$. Since, *ReLU* is a monotonic function and $x_{s'} > 0$ by the second constraint, we have $\sigma(x_{s'}) = x_{s'}$ and hence, $\sigma(\sum_{s \in S_i} W_i^l(s, s')x_s + b_i^l(s')) \leq x_{s'} \leq \sigma(\sum_{s \in S_i} W_i^u(s, s')x_s + b_i^u(s'))$. Hence, $x_{s'} = \sigma(\sum_{s \in S_i} w_{s,s'}x_s + b_{s'})$ for some $W_i^l(s, s') \leq w_{s,s'} \leq W_i^u(s, s')$ and $b_i^l(s') \leq b_{s'} \leq b_i^u(s')$. Hence, $v'(s')$ is obtained from $v$ using the definition of $[\![T]\!]_i$.

Case $z(q_{s'}) = 1$: In this case, $x_{s'} = 0$ and $\sum_{s \in S_i} W_i^l(s, s')x_s + b_i^l(s') \leq x_{s'} = 0$. Therefore $\sigma(\sum_{s \in S_i} W_i^l(s, s')x_s + b_i^l(s')) = 0 = x_{s'}$, therefore $v'(s')$ is obtained from $v$ using the definition of $[\![T]\!]_i$.