[Reviews · NeurIPS 2019]

Reviewer 1



Pros: 1. Paper is well written and easily readable. 2. It presents a novel approach for state space reduction of neural network that could be extended to similar problems. 3. It address an important issue of computational complexity in output range analysis for neural network. Cons: 1. This paper should explore different partitioning schemes and provide a timing comparison among them. 2. It should provide a metric to compare degree of over approximation as compared with approach in [4]

Reviewer 2



First of all, my knowledge of formal verification of neural networks is very limited, and I apologize for the limitations this poses on my review. That said, I found this paper very interesting, well written, and from my limited understanding of the literature, this seems like a novel and highly useful tool in the toolbox for verifying neural network models. I am strongly in favor of acceptance. My main questions are the following: * It is not clear to me what increase in false positives does the method introduce by relaxing the estimate of the output of the network to a superset. * I would like to see a more formal definition of the algorithm with the "moving pieces" (e.g. partitioning strategies) stated more explicitly. Then I would like to have a discussion of the considerations that go into defining these "moving pieces". * What are the practical limitations of the method on real-world network sizes and architectures.

Reviewer 3



Overall, I like the approach in the paper and the theoretical background looks solid (though I didn't check the proofs). My main problem with a paper is in the experimental part: * just one experiment is considered on rather toy neural network * no comparison with other methods is made Thus, it is impossible to evaluate the usefulness of the proposed method.

[Author Response · NeurIPS 2019]

We thank the reviewers for their encouraging feedback, and suggestions for improvement.

The main contribution of the paper is a novel abstraction technique to reduce the state-space for output range analysis of neural networks. Our approach is orthogonal to existing approaches in that it can be used in conjunction with existing verification approaches. In particular, it allows one to apply the existing approaches on a "smaller" system with fewer neurons, though the verification techniques need to be extended from the model of neural networks to those with "interval weights". In the paper, we demonstrate that such an extension is feasible, by applying it to a verification technique based on MILP encoding. Next, we address the issues raised by the reviewers:

1. Just one experiment, no comparison with existing methods. What are the practical limitations of the method on real-world network sizes and architectures? (Reviewer 2 and Reviewer 3)

   We would like to note that the area of formal verification for neural network is in its nascent stages, and extensive benchmarking has not been performed for the same. Hence, we performed our experimentation on a well known example used in the related works, to show a proof of concept for our work. We agree that more extensive experimentation is required to fully understand the scalability of the approach. Our experiments focused on evaluating the benefits of the abstraction technique, and extensive comparisons with other techniques were not performed, since, our methods is orthogonal to existing verification methods. However, implicitly we compare our method with the MILP based encoding method, for instance, in [4], where our experimental analysis shows that the abstraction + MILP outperforms just MILP based encoding. We envision that several heuristics will need to be explored to scale this and other verification techniques to the real-world network sizes. It will be interesting to explore the extension of our methods for different architectures.

2. Degree of approximation and false positives. (Reviewer 1 and Reviewer 2)

   Figure 10 briefly highlights the trade-off between the degrees of approximation and the output range. In general, as the number of abstract nodes increases, the approximation error of the interval neural network decreases, and the output range becomes more precise. However, we observed that the exact output range might be affected by the particular partition, even when we fix the number of abstract nodes. Hence, it is important to explore partitioning strategies.

3. Exploration of partitioning strategies and comparison. (Reviewer 1 and Reviewer 2)

   This is an important line of research, and is known in the formal methods community as "abstraction/refinement strategies", which refers to iterative constructing more precise abstraction starting from some abstraction. One such methodology is known as counter-example guided abstraction refinement, wherein, a "counter-example" from the current abstraction is analyzed to construct the next abstraction. We intend to explore the idea in the near future, however, it requires new foundations and insights, and will require more extensive research. In this paper, we have explore random refinement strategies to highlight the need for clever refinement strategies.

4. Formal definition of algorithms. (Reviewer 2)

   We will add an algorithm summarizing the abstraction and verification procedures.

[Meta-Review · NeurIPS 2019]

The paper presents an interesting contribution to deep nets verification. While the paper tackles a significant problem, a better reference to previous work will certainly improve this submission. Particularly, it will be nice to emphasize the importance of the abstraction, both practically and theoretically.